# Structural mechanisms for binding and activation of a contact-quenched fluorophore by RhoBAST

Yufan Zhang[1,5], Zhonghe Xu ®[2,5], Yu Xiao[2,5], Haodong Jiang[3,5], Xiaobing Zuo[4], Xing Li ®[3] ✉ & Xianyang Fang ®[1,2] ✉

The fluorescent light-up aptamer RhoBAST, which binds and activates the fluorophore–quencher conjugate tetramethylrhodamine-dinitroaniline with high affinity, super high brightness, remarkable photostability, and fast exchange kinetics, exhibits excellent performance in super-resolution RNA imaging. Here we determine the co-crystal structure of RhoBAST in complex with tetramethylrhodamine-dinitroaniline to elucidate the molecular basis for ligand binding and fluorescence activation. The structure exhibits an asymmetric "A"-like architecture for RhoBAST with a semi-open binding pocket harboring the xanthene of tetramethylrhodamine at the tip, while the dinitroaniline quencher stacks over the phenyl of tetramethylrhodamine instead of being fully released. Molecular dynamics simulations show highly heterogeneous conformational ensembles with the contact-but-unstacked fluorophore–quencher conformation for both free and bound tetramethylrhodamine-dinitroaniline being predominant. The simulations also show that, upon RNA binding, the fraction of xanthene-dinitroaniline stacked conformation significantly decreases in free tetramethylrhodamine-dinitroaniline. This highlights the importance of releasing dinitroaniline from xanthene tetramethylrhodamine to unquench the RhoBAST–tetramethylrhodamine-dinitroaniline complex. Using SAXS and ITC, we characterized the magnesium dependency of the folding and binding mode of RhoBAST in solution and indicated its strong structural robustness. The structures and binding modes of relevant fluorescent light-up aptamers are compared, providing mechanistic insights for rational design and optimization of this important fluorescent light-up aptamer-ligand system.

RNAs play essential roles in multiple cellular processes, such as gene expression, processing, catalysis, and gene regulation[1]. To gain insights into the complex life of RNAs, it's crucial to image and track RNAs of interest (ROI) in living cells with high spatial and temporal resolution using state-of-the-art fluorescence microscopy[2–4]. The prerequisite for fluorescence visualization is efficient, specific, and robust labeling of the biomolecules of interest with fluorescent tags[5,6]. While enormous progress has been made in the past to develop the naturally

[1]Key Laboratory of RNA Science and Engineering, Institute of Biophysics Chinese Academy of Sciences, Beijing, China. [2]Beijing Frontier Research Center for Biological Structure, School of Life Sciences, Tsinghua University, Beijing, China. [3]Institute of Zoology, Beijing Institutes of Life Science, Chinese Academy of Sciences, Beijing, China. [4]X-ray Science Division, Argonne National Laboratory, Lemont, IL, USA. [5]These authors contributed equally: Yufan Zhang, Zhonghe Xu, Yu Xiao, Haodong Jiang. ✉e-mail: lix@biols.ac.cn; fangxy@ibp.ac.cn

occurring fluorescent proteins (FPs) into valuable fluorescent tagging tools, enabling spatial and dynamic studies of the proteome and cellular state[7–9], the lack of intrinsically fluorescent RNA counterparts has precluded the study of transcriptome as well as RNA-dependent processes in vitro and in vivo[10].

A burgeoning field of fluorescent light-up RNA aptamers (FLAP) has emerged as RNA counterparts to FPs for RNA detection and imaging in living cells[11–13]. FLAPs are in vitro selected RNA sequences that bind and dramatically enhance the fluorescence of fluorogenic ligands which are cell-permeable and exogeneous to the biological system. Currently, three general mechanisms have been exploited by FLAPs to activate different types of fluorogenic ligands: twisted intramolecular charge transfer (TICT)[14], contact quenching (CQ)[15,16], and spirolactonization (SP)[17], leading to a high diversity of fluorescence and binding properties. To date, more than 70 FLAP-ligand systems have been generated[14]. The majority of FLAPs, including the earliest MG aptamer[18] and SRB aptamers[19], and the later ones such as spinach[20,21], broccoli[22], corn[23], mango[24], chili[25,26], and pepper[27], are developed based on the TICT mechanism. FLAPs can be genetically fused to the target ROI. Due to their high brightness, better programmability, and smaller sizes, they are increasingly regarded as promising tools for RNA detection and imaging in living cells.

Recently, a series of FLAPs including the RhoBAST[28], biRhoBAST[29], o-Coral[30], and SiRA[31], gained much attention because of their extraordinary performance in super-resolution RNA imaging. These FLAPs target the derivatives of rhodamine, an intrinsically fluorescent dye known for extraordinary photostability, brightness, and good membrane permeability. RhoBAST was specifically optimized from its parental sequence, SRB-2, which was originally selected against sulforhodamine B. biRhoBAST and o-Coral can be regarded as dimeric variants of RhoBAST and SRB-2, respectively. It is suggested that RhoBAST could activate different rhodamine-based ligands, such as the tetramethylrhodamine-dinitroaniline (TMR-DN) and spirocyclic rhodamine (SpyRho)[32] (Fig. 1a), via the CQ and SP mechanisms, respectively, which display super high brightness, high binding affinity ($K_D$ ~ 30 nM), enhanced photostability and faster exchange kinetics than other FLAP systems under the same condition[28,32]. Additionally, RhoBAST displays high thermostability ($T_m$ = 79 °C), independent of sodium or potassium ions but strictly dependent on $Mg^{2+}$ (0.25 mM) for folding, indicating that the RhoBAST aptamer is robust and can function optimally under physiological conditions.

To understand the molecular basis for high-affinity binding and fluorophore activation by RhoBAST, we determined the cocrystal structure of RhoBAST in complex with TMR-DN, a fluorophore–quencher conjugate. RhoBAST adopts an asymmetrical "A"-like architecture organized with a four-way rather than the predicted three-way junction. The topology is stabilized by extensive tertiary interaction networks between the loops L3 and L4, harboring a semi-open ligand binding pocket at the tip of "A"-like structure. Importantly, the crystal structure reveals an incompletely unquenched conformation for TMR-DN bound to RhoBAST, of which the xanthene moiety of TMR is snugly fitted into the semi-open binding pocket, whereas the DN quencher is stacked over the phenyl ring rather than releasing from TMR. Molecular dynamics (MD) simulations suggest highly heterogeneous conformational ensembles with predominant contact–but–unstacked fluorophore–quencher conformation for both free and bound TMR-DN, along with a significant decrease of fraction of xanthene-DN stacked conformation in free TMR-DN upon RNA

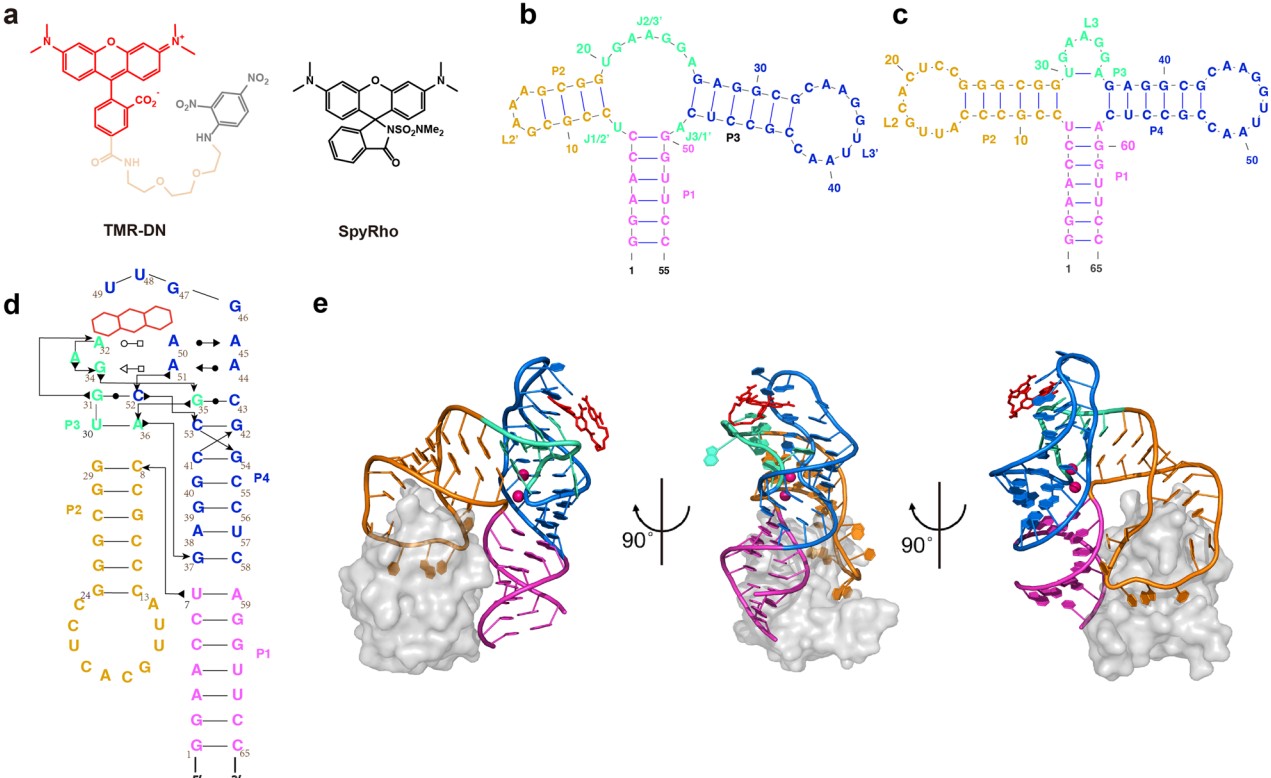

**Fig. 1 | Overall structure of the TMR-DN-bound RhoBAST aptamer. a** Chemical structures of the conditional fluorophores of TMR-DN and SpyRho. **b** The previously predicted secondary structure of RhoBAST WT. **c** An updated secondary structure of RhoBAST (L2-U1 construct) from the crystal structure in this work. **d** A schematic representation of the secondary structure of RhoBAST highlighting long-range interactions and non-canonical base pairs. Thin lines with arrowheads and Leontis–Westhof symbols denote connectivity and base pairs, respectively. **e** Cartoon representation of the three-dimensional structure of the RhoBAST aptamer in complex with TMR-DN (red stick representation) and U1A protein (gray surface representation) in three orthogonal views. Two $Mg^{2+}$ ions are shown as magenta.

binding. Using small-angle X-ray scattering (SAXS) and isothermal titration calorimetry (ITC), we characterized RhoBAST in solution, revealing that Mg²⁺ is strictly required for folding and pre-organization of the binding pocket for efficient ligand binding. Together, these results provide mechanistic insights into the recognition and activation of a contact-quenched fluorophore by RhoBAST, representing the starting point for rational design and optimization of this important FLAP-ligand system.

## Results

### Constructs and crystallization of RhoBAST
The predicted secondary structure of RhoBAST consists of a three-way junction connected by two loops: L2′ and L3′ (Fig. 1b)[28]. To facilitate crystallization and phasing, the L2′ or L3′ of RhoBAST were initially replaced with the U1 RNA hairpin loop, resulting in two constructs of L2′-U1 and L3′-U1, respectively (Supplementary Fig. 1a). The human RNA-binding protein U1A, which specifically binds to the U1 RNA loop with high affinity, has previously been established as co-crystallization chaperone and successfully used for solving the structures of several RNAs[33]. While the L2′-U1 construct retains the same high affinity as wild-type RhoBAST binding to TMR-DN, L3′-U1 doesn't bind to TMR-DN at all (Supplementary Fig. 1b), indicating that L3′ but not L2′ loop is essential to the overall folding and ligand binding of RhoBAST. These results are consistent with previous conservation analysis on SRB-2, the predecessor of RhoBAST, that the L2′ is non-conserved, but the highly conserved regions of J2/3′, L3′ and J3/1′ may interact with each other to create the sulforhodamine-binding site[19]. The L2′-U1 construct binds the U1A protein with a similar high affinity ($K_D = 7.94 \pm 1.74$ nM) as that reported for the U1 RNA loop[34] (Supplementary Fig. 1c), we thus proceeded with this construct for crystallization.

In co-crystallization with the U1A protein, the crystal structure of the L2′-U1 construct in complex with TMR-DN was determined at 2.75 Å resolution, and the data collection and refinement statistics are summarized in Supplementary Table 1. The space group is $I2_12_12_1$, and each asymmetrical unit contains three complexes. The initial phases were obtained by molecular replacement using U1A coordinates (PDB: 5DDR) as a search model[35]. The three complexes are of high structural similarity (Supplementary Fig. 2). However, the electron density for the DN moiety of TMR-DN is missing in two of them, likely due to the flexibility of the linker connecting the fluorophore and quencher. We therefore focus our structural analysis on the complex in which the electron density of the ligand TMR-DN is complete and visible.

### Overall structure of RhoBAST bound to TMR-DN
Interestingly, the RhoBAST folds into an asymmetric "A"-like architecture, which is organized with a four-way rather than the previously predicted three-way junction consisting of three A-form helices (updated as P1, P2, and P4) and one short pseudo-helix P3 connected by three loops (updated as L2, L3, and L4) (Fig. 1c–e). The pseudo-helix P3 is established by one Watson–Crick (WC) base pair between two residues (U30–A36) within the previously proposed junction J2/3′. As observed in a typical family-H RNA four-way junction[36], the four helices of RhoBAST, P1 with P4, and P2 with P3, are coaxially stacked, forming two superhelices $H_{41}$ and $H_{23}$ that are roughly antiparallel to each other. The L3 and L4 loops interact extensively (Fig. 1d), which stabilizes the "A"-like architecture and creates a binding platform for the fluorophore ligand at the tip of the "A"-like structure (Fig. 1e). The U1A crystallization chaperone protein is found to bind with both legs of the "A"-like structure, either through specific interactions with the U1A loop adjoining to stem P2 or nonspecific interactions with the duplexes of P1 and P2 such as salt bridges as well as hydrogen bonds (Supplementary Fig. 3).

### Loop–loop interactions
The L3 and L4 loops containing five and ten nucleotides, respectively, are anchored through intricate tertiary interaction networks

comprising two base pairings, two base triples, coaxial stacking interactions and Mg²⁺ (Fig. 2a). The nucleobases of G35 within L3 and C52 within L4 flip out from their residing loops and position into the major grooves of their counterparts (i.e., L4 and L3, respectively), forming two interwoven canonical WC base pairs, G35–C43 and G31–C52, respectively (Fig. 2b). Notably, the 4-amino of C52 forms two hydrogen bonds with the phosphate oxygen and N7 of its preceding residue A51, the 2′-OH of C52 with the N6 and N7 of A36, the O3′ and O4′ of C52 with the 2′-OH of G35, are also involved in hydrogen bonding interactions (Fig. 2c, d). These interactions may not only favor the conformation of C52 and G35 flipping off their residing loops, therefore stabilizing the interwoven WC base pairing, but also strengthen the stacking of the WC base pairs with the adjoining superhelices (G31–C52 with $H_{23}$, G35–C43 with $H_{41}$) and the two consecutive non-canonical base triples (A32•A50•A45, G34•A51•A44) (Fig. 2e). Although A36 from helices-interface of P3 and C52 flipped off loop4 form canonical base pairing with their complementary bases, interestingly, they also adopt C2′-endo sugar pucker, which may facilitate reverse the direction of the following nucleotide and entry into loop 3, respectively, and to form co-stacking of A36 and C52.

Of the two base triples, the nucleobases of central residues (A50 and A51) within the L4 loop on one side form non-canonical base pairs with residues of A32 and G34 within the L3 loop through *trans*-WC-Hoogsteen (A32•A50) base pairing and *trans*-Hoogsteen-sugar edge (A51•G34) base pairing (Fig. 2f, g), respectively, and on the other side interact with residues of A45 and A44 through A-minor interactions. The backbones between dinucleotides A50–A51 and A44–A45 in anti-parallel direction are held together by ribose zipper involving base-ribose and ribose-ribose hydrogen bonding interactions (Fig. 2e). It's worth noting that the tetranucleotides from G31 to G34 (³¹GAA³⁴G) within L3 fold into an intercalated conformation, of which G34 is stacked and sandwiched above G31 and below A32, and A33 reverses its backbone direction from upward to downward and extrudes its nucleobase to solvent (Fig. 2h). Except, for G31, sugars of all the other nucleotides (A32, A33, G34) adopt a C2′-endo conformation, which may be favorable for its intercalated folding and the formation of two base-triples with L4 loop. Within each base triple, the nucleobases are almost coplanar with each other, especially for the base pairs involved in major groove interaction. Between the two base triples, they are nearly parallel to each other, which is highly favored for purine stacking, in addition, the amino and imino of G34 within G34•A51•A44 are also found to make hydrogen bonds with phosphate oxygen atoms of A50 within A32•A50•A45, further improving the stacking stability, thus the two consecutive base triples could play an essential role in the interlocking between loops L3 and L4 (Fig. 2i).

Furthermore, two Mg²⁺ ions can be identified to cooperatively coordinate with two couples of adjacent phosphate moieties around the helices interfaces, such as C8-Mg²⁺₁-G37 (holding P2 and P4) and A36-Mg²⁺₂-C53 (holding P3 and P4) (Fig. 2j). The Mg²⁺ ions shield repulsive negative charges on the RNA backbone and serve as clamps to hold these helices together, setting up a base anchor for L3 and L4 loop–loop interactions.

### Structure of the fluorophore binding core and RhoBAST-bound TMR-DN
The experimental electron density map allowed complete tracing of not only the RNA RhoBAST but also the fluorophore–quencher conjugate TMR-DN, of which the DN quencher is tethered to the phenyl ring of the fluorophore TMR via a flexible polyethylene glycol linker ($(PEG)_n$, $n = 2$)[28] (Fig. 3a). The fluorophore binding core residing at the tip of the "A" architecture was created by the 4-nt apex of loop L4 (G46–G47–U48–U49, hereafter termed capping loop) along with the A32•A50•A45 base triple underneath, resembling a half-closed rectangular box (Fig. 3b). Distinguished from the two base triples, the nucleobases within the capping loop are not co-planar to each other.

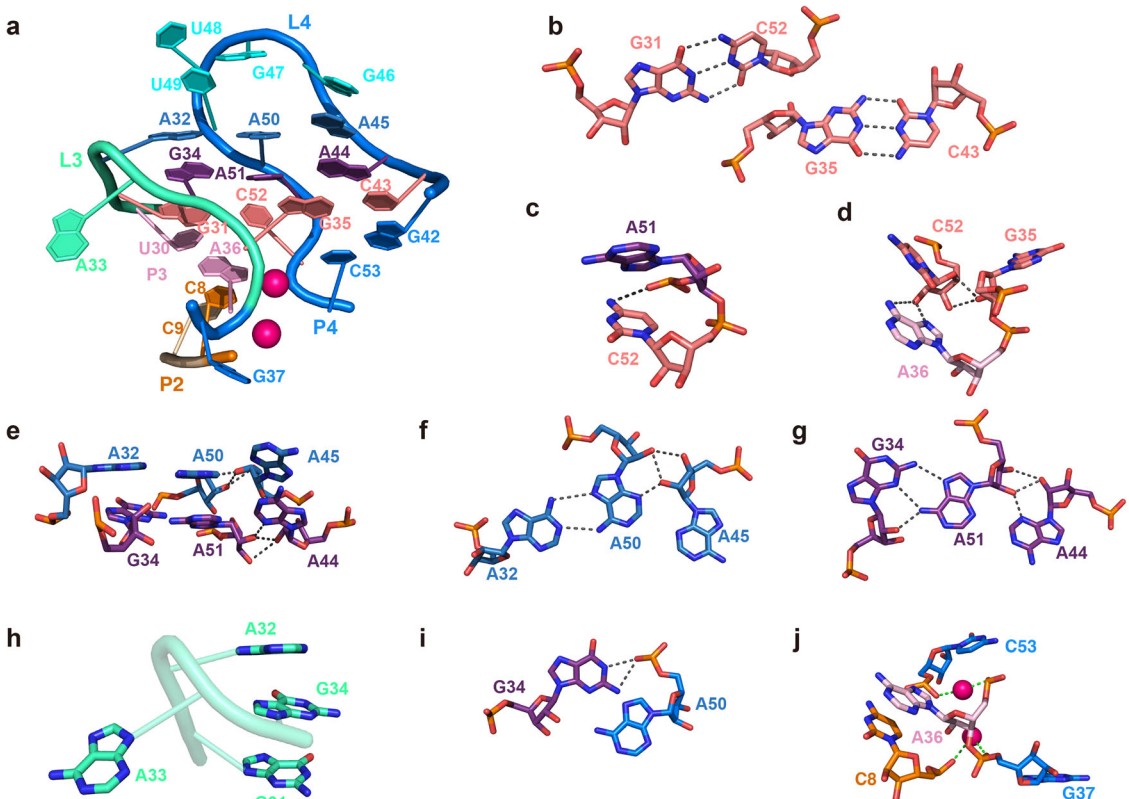

**Fig. 2 | Loop–loop interactions. a** Extensive tertiary interaction networks exist between loops L3 and L4. **b** Two interwoven WC base pairs between L3 and L4. **c**, **d** Tertiary interactions involved in C52 and its neighboring residues. **e** Two stacked base triple interactions and ribose zipper. **f**, **g** Diagram of interaction networks of the two consecutive base triples. **h** Intercalated conformation of the tetranucleotides from G31 to G34 within L3. **i** Tertiary interactions involved in G34 and A50. **j** Two Mg²⁺ (magenta sphere) bring together residues C8 and G37, A36 and C53, respectively. The gray and green dashed lines denote the hydrogen bonds and the coordination of Mg²⁺ ions with the RNA phosphate groups, respectively.

Among them, only G46 is stacked with A45 within the A32•A50•A45 base triple, and the other nucleotides adopt a C2′-endo sugar conformation (Fig. 3c), which is likely favorable to reversing the backbone direction and formation of tertiary interactions. The relative orientation of G47 nucleobase is stabilized by one hydrogen bonding between N7 of G47 and O2′ of G46, and an inclined base–base interaction comprising two hydrogen bonding between the O4 and N3 of U49 and N3 and N2 of G47, respectively (Fig. 3d). The nucleobase of U48 is extruded out of the capping loop and makes no contact with other nucleotides. Except for the base–base interactions with G47, the O2′ of U49 forms two hydrogen bonds with N6 of A32 and O6 of G34, respectively (Fig. 3e).

RhoBAST-bound TMR-DN has been presumed to adopt an unquenched conformation and the quencher DN will be released from the fluorophore TMR of TMR-DN alone upon binding to RhoBAST[14,28], resulting in a significant fluorescence increase. From the crystal structure, the xanthene moiety of the fluorophore TMR intercalates into the binding core in a planar conformation with its long axis approximately parallel to the direction of base pairing between A32 and A50, which is predominantly sandwiched below G47 nucleobase and above nucleobases of A32 and A50, resulting in continuous stacking along with the capping loop, the base triples and the super helices of H₂₃ and H₄₁ that might further stabilize the ligand–RNA interactions (Fig. 3c). However, the encapsulation of xanthene moiety into the core only buries 48% of TMR solvent accessible surface area (SASA), noticeably lower than that of other RNA aptamers which are normally around 60–70%[26,37–49](Supplementary Fig. 4 and Supplementary Table 2). Outside of the binding core, the phenyl ring of TMR adopts an inclined orientation (~60°) to the tricyclic ring of xanthene

(Fig. 3f), the ortho-carboxylate group within the phenyl ring forms two hydrogen bonds with the amino and imino groups of G47, respectively (Fig. 3g). However, rather than completely releasing from the xanthene moiety, the quencher DN is stacking over the phenyl ring of TMR and making a hydrogen bond with the 2-amino of G47 (Fig. 3f, g). Thus, these stacking and hydrogen bonding interactions might constrain the distance between the xanthene moiety and the DN quencher, and RhoBAST-bound TMR-DN exhibits an incompletely unquenched conformation, in line with the modest fluorescence quantum yield of 0.57 as reported[32].

### Enhanced-sampling simulations of TMR-DN alone and in complex with RhoBAST

The fluorescence quantum yields, $\Phi_F$, of free TMR-DN ($\Phi_F = 0.08$), RhoBAST-bound TMR-DN ($\Phi_F = 0.57$), and RhoBAST-bound quencher-free TMR probe ($\Phi_F = 0.92$) suggest that the quenching of TMR by DN is highly efficient in free TMR-DN, but the unquenching of TMR upon RhoBAST binding is incomplete[15,32]. To better understand the activation mechanism of TMR-DN by RhoBAST, we performed enhanced sampling MD simulations to explore the conformational landscapes of free and RhoBAST-bound TMR-DN. We employed a hybrid replica exchange with the Hamiltonian and Temperature method to accelerate energy barrier crossing[50]. The conformers from the replica running at the lowest temperature were used for analysis. The spatial distributions of DN around TMR of free and RNA-bound TMR-DN were visualized in Fig. 4a, b. The wide distributions indicate that TMR-DN alone or in complex with RhoBAST samples a highly heterogeneous conformational ensemble. In the ensemble of TMR-DN alone (Fig. 4a), DN prefers to contact with the TMR moiety with higher occupancy

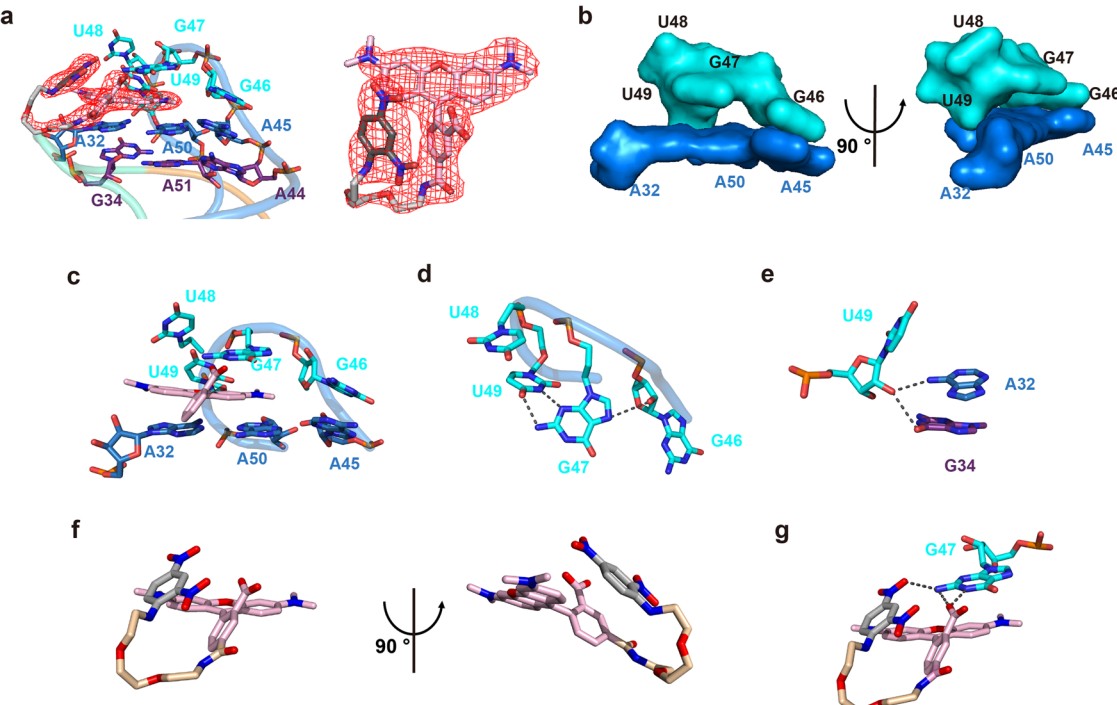

**Fig. 3 | Structure of the fluorophore-binding core and the ligand TMR-DN.**
**a** Crystal structure of the ligand TMR-DN intercalated into the fluorophore-binding core. The red mesh grid depicts the 2|Fo|−|Fc| electron density map of the ligand (contour level: sigma = 1.0). **b** Surface representation of semi-open binding pocket formed with the capping loop and the underneath A32•A50•A45 base triple in two views. **c** The fluorophore TMR is sandwiched below the G47 nucleobase and above nucleobases of A32 and A50. **d** G47 nucleobase forms hydrogen bonds with the G46 sugar ring and U49 nucleobase. **e** The O2´ of U49 forms two hydrogen bonds with N6 of A32 and O6 of G34, respectively. **f** Stick representation of the RhoBAST-bound TMR-DN ligand (RhoBAST not shown) in two views. The phenyl ring of TMR adopts an inclined orientation (-60°) to the tricyclic ring of xanthene, which is partially stacked with the quencher DN. **g** The DN quencher and the ortho-carboxylate group within the phenyl ring form one and two hydrogen bonds with the nucleobase of G47, respectively.

density on the surface around the xanthene than the phenyl moiety. Upon binding to RhoBAST (Fig. 4b), the contact of DN with xanthene is highly suppressed.

To obtain quantitative comparisons of the conformational ensembles of TMR-DN alone and in bound form, we analyzed the contact number ($Q$) and centroid distance ($d$) between xanthene or phenyl ring moieties of TMR fluorophore and DN quencher (Fig. 4c–h and Supplementary Table 3). The contact number ($Q$) between the TMR fluorophore or its xanthene or phenyl ring and the DN quencher in each conformer, which describes non-covalent intramolecular interactions based on interatomic distances, were calculated and termed as $Q_{Tq}$ (the sum of $Q_{xq}$ and $Q_{pq}$), $Q_{xq}$, and $Q_{pq}$, respectively. Similarly, the centroid distance between TMR xanthene or phenyl ring and DN quencher was defined as $d_{xq}$ and $d_{pq}$, respectively. Consistent with the analysis of spatial distributions (Fig. 4a, b), the binding of RhoBAST to TMR-DN leads to a remarkable decrease in $Q_{xq}$ and $Q_{Tq}$ (Fig. 4c, d), accompanied by an apparent increase in $Q_{pq}$ (Fig. 4e). If using a $Q_{Tq}$ of 5 as the cut-off, all the conformers can be categorized into the contact ($Q_{Tq} \geq 5$) and the non-contact ($Q_{Tq} < 5$) groups, and the population of the non-contact conformers TMR-DN increases from 10% to 40% upon binding to RhoBAST. Similarly, the changes in distance distributions between xanthene and DN ($d_{xq}$) and between phenyl ring and DN ($d_{pq}$) of TMR-DN from free to bound forms further confirm that binding of TMR-DN to RhoBAST causes the dissociation of DN from xanthene, as evidenced by increased populations of longer distances (Fig. 4f, g). The larger population of contact conformation in TMR-DN alone is expected to efficiently quench its fluorescence, consistent with the reported low quantum yield of TMR-DN alone[32].

Based on the stacking property (see Supplementary Methods and Supplementary Fig. 10), all the contact conformers can be further classified into three subgroups, DN-xanthene stacked (StackX), DN-

phenyl stacked (StackP), and contact-but-unstacked (contact-unstacked), the populations of each subgroup for free and bound TMR-DN are summarized in Fig. 4h. The representative conformers for each subgroup are visualized in Fig. 4i, j (Supplementary Data 1). For TMR-DN alone, single-point energy calculations using density functional theory indicate that the three subgroups of contact conformers share comparable energies, which are more energetically favorable than those in the non-contact group (Supplementary Table 4). However, the contact-unstacked conformation rather than the StackX or StackP conformation is predominant in both free and bound TMR-DN. The StackX confirmation only accounts for around 28% of all conformers of TMR-DN alone, remarkably less than that of contact-unstacked conformation (58%). It is likely that the contact-unstacked conformation is beneficial to retain the contact-quenched mechanism and high-affinity RNA binding, as the StackX conformers may need more energy to release the DN from xanthene than the contact-unstacked ones. In line with our hypothesis, TMR is reported to bind RhoBAST with a similar affinity to TMR-DN-bound RhoBAST[32]. Upon binding to RhoBAST, xanthene becomes less accessible to DN, as demonstrated by the contact number analysis for each atom in TMR moiety (Fig. 4c, d and Supplementary Fig. 5a). Consequently, the population of StackX conformers significantly decreases from 28% to 1% (Fig. 4h). Such population shift is roughly in line with the change in quantum yield of TMR-DN upon binding to RNA. By contrast, the population of StackY conformers slightly increases from 2% to 6% (Fig. 4h). The three subgroups of contact conformers along with the non-contact group of conformers of free and bound TMR-DN were also mapped to 2D scatter plots of $Q_{xq}$ vs $Q_{pq}$ and $d_{xq}$ vs $d_{pq}$ (Supplementary Fig. 5b–e). Consistently, the wide distributions indicate that both free and bound TMR-DN samples are highly heterogeneous conformational ensembles. The highly heterogeneous conformational

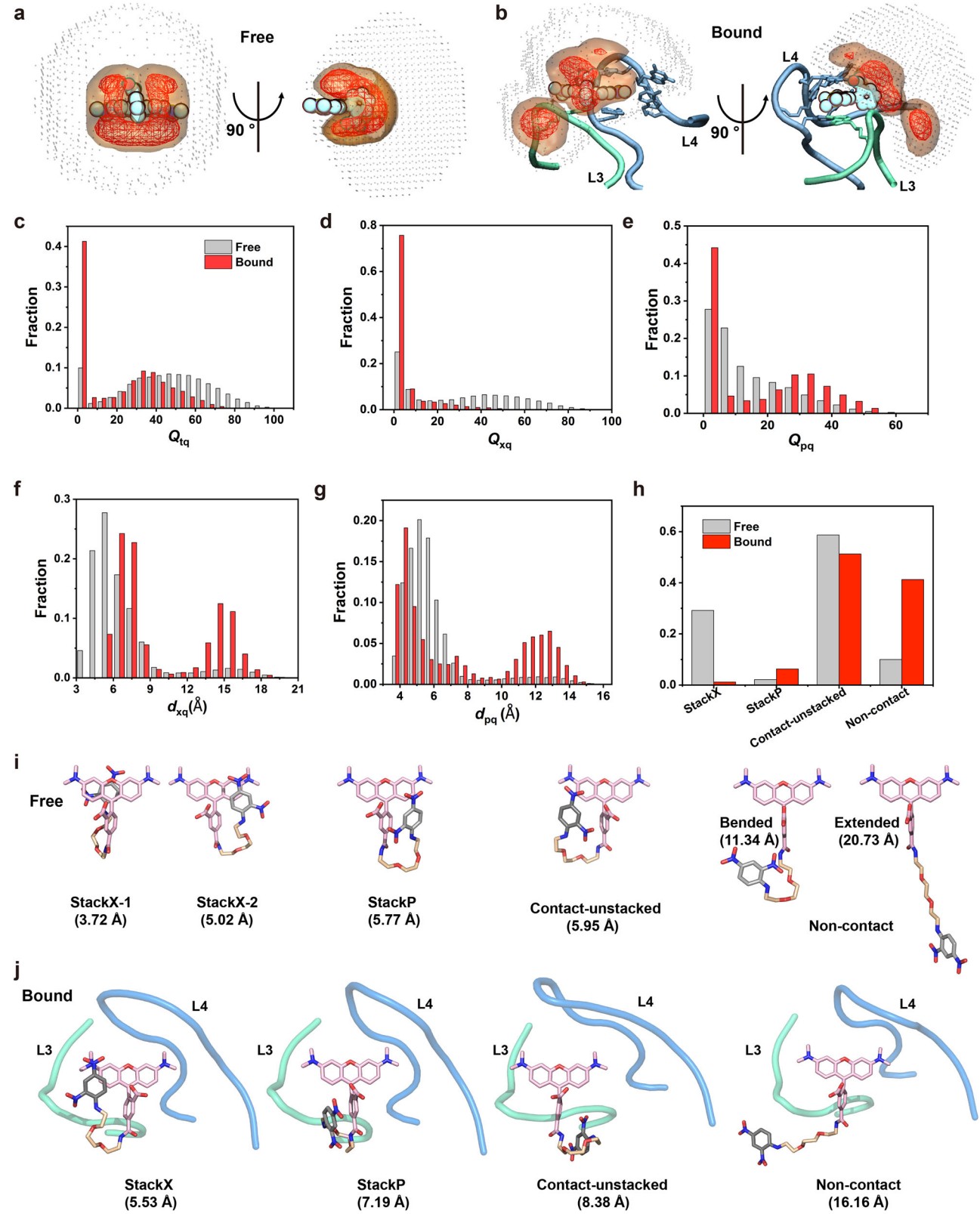

ensemble with predominant contact-unstacked conformation may explain the incomplete unquenching and modest quantum yield ($\Phi_F = 0.57$) of the RhoBAST−TMR-DN complex.

## Mutational analysis of the RhoBAST

To examine the importance of tertiary interaction networks in Rho-BAST, we generated site-directed mutants of the wild-type RNA construct (Fig. 5a). Their ability to bind TMR-DN and activate TMR-DN fluorescence were evaluated using ITC and fluorescence enhancement assay, respectively (Fig. 5b, c, Supplementary Fig. 6, and Supplementary Table 5). We examined the nucleotides involved in the interwoven WC base pairs. Two point mutants (G31C, C52G) and a double mutant (G35C/C52G), which are assumed to disrupt either or both of the interwoven WC base pairs, all abolished their ability to bind with

**Fig. 4 | Conformational landscapes of free and RhoBAST-bound TMR-DN.**
**a**, **b** Spatial distribution analysis of the DN quencher around the TMR moiety of TMR-DN alone (**a**) or in bound form (**b**). The red mesh, orange surface, and gray dot represent the isovalue of 10%, 1%, and 0.01% with respect to its highest density, respectively. In the complex system, only partial L3 (cyan) and L4 (blue) are shown as cartoon representations for clarity. The nucleobases involved in interaction with TMR-DN are displayed as stick representations. **c–e** The distributions of contact numbers ($Q$) between the DN quencher and TMR fluorophore (**c**) or the xanthene (**d**) or phenyl ring (**e**) moieties. **f**, **g** The distributions of centroid distances ($d$)

between the DN quencher and the xanthene (**f**) or phenyl ring (**g**). **h** The fraction for each group of conformers including the DN-xanthene stacked, DN-phenyl stacked, contact-but-unstacked, and non-contact for both systems of TMR-DN alone and in complex with RNA. **i**, **j** Representative conformers of each subgroup of the ensembles of TMR-DN alone (**i**) or in complex with RhoBAST (**j**). The centroid distances between xanthene and DN in the representative conformers are indicated. For clarity, only a portion of the RNA was shown in the RhoBAST–TMR-DN complex. Source data for panels **c–i** are provided as a Source data file.

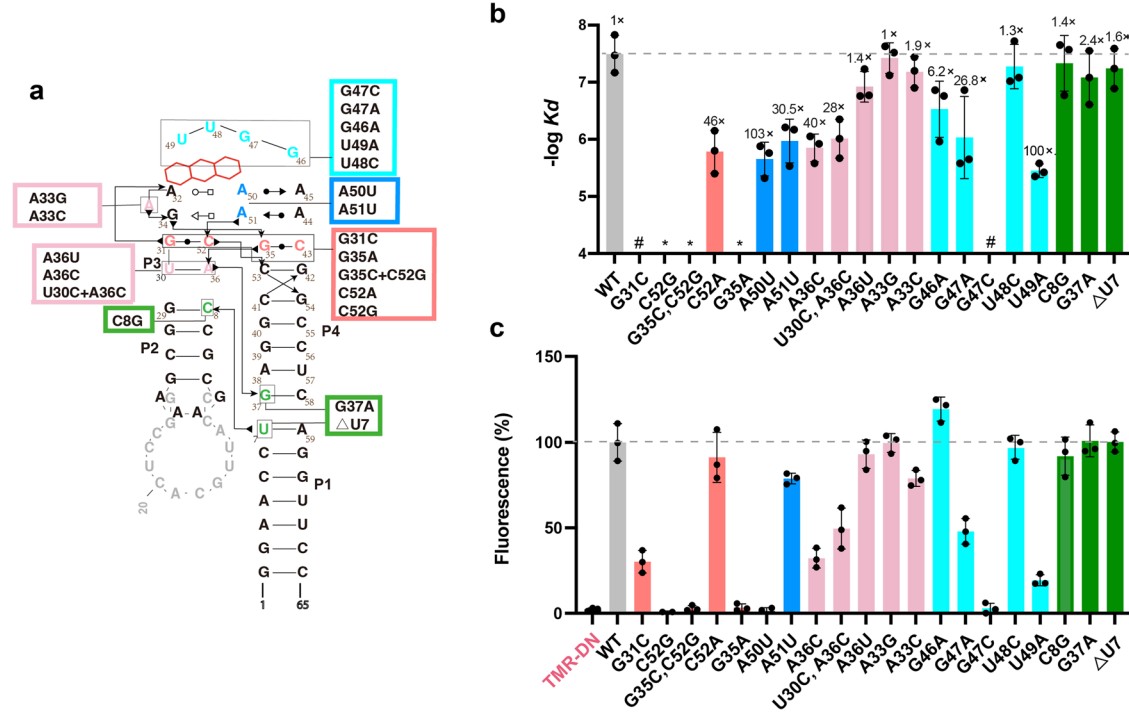

**Fig. 5 | Mutational effects on ligand binding and fluorescence enhancement by RhoBAST. a** Mutants of RhoBAST WT presumed to disrupt certain tertiary interactions are mapped to its secondary structure. Residues are numbered corresponding to the L2-U1 crystallization construct, and secondary structures are shown in gray for comparison. **b** ITC analysis of ligand binding affinity of all the

mutants. ("*"means no binding; "#" means weak binding) (mean ± s.d., $n = 3$). **c** Fluorescence of all the RhoBAST mutants in the presence of TMR-DN are normalized to the WT (mean ± s.d., $n = 3$). Source data for panels **b–c** are provided as a Source data file.

TMR-DN (Fig. 5b). It's likely the G31C and C52G point mutants form a local base pair within loops L3 (C31'–G35) and L4 (C43–G52'), respectively, and the double mutant forms two local base pairs within the loops L3 (G31–C35') and L4 (C43–G52'), thus neither of the original two interwoven WC base pairs is preserved in all these mutants (Supplementary Fig. 7b–d). As a result, the loop–loop interactions between L3 and L4 are adversely affected and the "A" architecture as well as the ligand binding pocket is disrupted, leading to the complete loss of ligand binding. Interestingly, a mismatched mutant (C52A) decreased its ligand binding affinity by 46-fold, but the mismatched G35A mutant totally lost its ligand binding activity. It's likely the C52A mutant retains but the G35A mutant disrupts the G35–C43 interwoven WC base pair, though the G31–C52 interwoven WC base pair may remain in the G35A mutant, implying that the G35–C43 interwoven base pair is more essential in maintaining the "A" architecture and ligand binding ability (Supplementary Fig. 7e, f). Compared to wild-type RhoBAST, neither the C52G nor G35A mutants have any obvious fluorescence enhancement activity towards the fluorophore TMD-DN, but the C52A mutant reduces fluorescence to 85% of the wild type (WT) (Fig. 5b). Their activity to enhance TMD-DN fluorescence roughly correlates with their ligand binding affinity, supporting the functional importance of these interaction networks. Taken together, the two interwoven WC base

pairs are essential to the binding and activation of TMD-DN by RhoBAST.

We next evaluated the point mutants (A50U, A51U) corresponding to mutations at the central residues of the two base triples (Fig. 5b, c). The ligand binding affinities of the two mutants were decreased by nearly 100 and 30 folds, respectively. Compared to the RhoBAST WT, the fluorescence enhancement activity of the A50U mutant on TMR-DN was not obvious, and the A51U mutant reduced fluorescence to 75% of the WT. The mutations might have destabilized the base triples and weakened their stacking with the ligand, leading to reduced binding affinity and significant loss or reduction of fluorescence activation. Thirdly, we assessed the nucleotides within the pseudo-helix P3 and loop L3. As P3 contains only one base pair (U30–A36) which might be more vulnerable, mutations within P3 (A36C, A36U, and U30C/A36C) are expected to destabilize the RNA architecture and damage ligand binding. The point mutant (A36C) and a double mutant (U30C/A36C) reduced TMR-DN binding affinities by 40 and 28 folds, respectively, by contrast, the ligand binding affinity of the A36U mutant is decreased by only 3.5 fold. As the A36U mutant has the potential to form a noncanonical U–U base pair with two hydrogen bonds, its structure and ligand binding ability may be minimally affected. Residue A33 within loop L3 makes no contact with other residues, unsurprisingly,

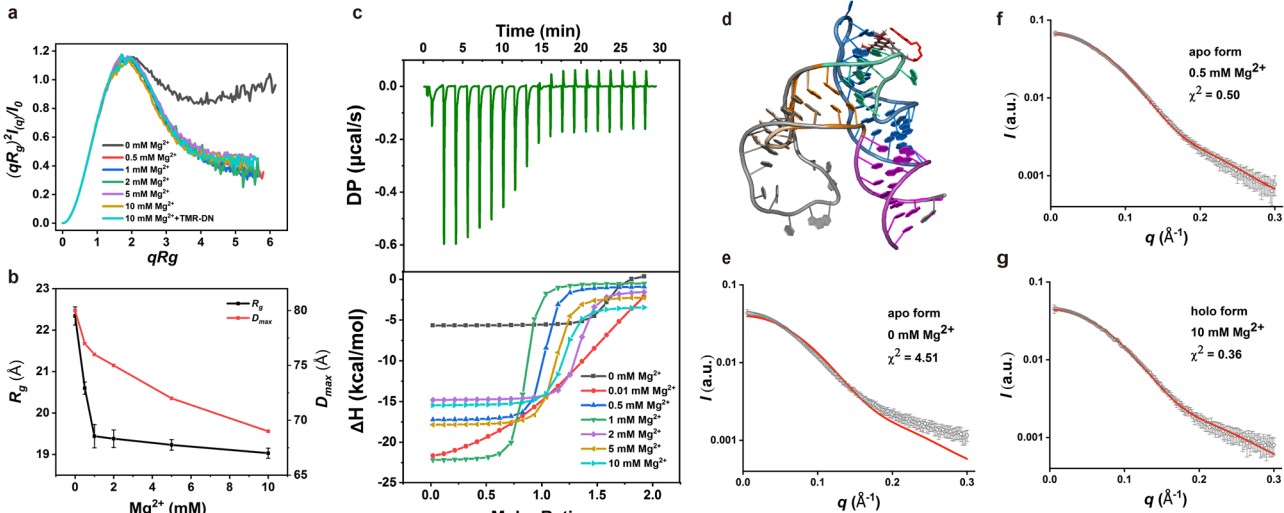

**Fig. 6 | SAXS and ITC analysis of RhoBAST folding and ligand binding.**
**a** Dimensionless Kratky plots of RhoBAST WT in the presence of various concentrations of $Mg^{2+}$ and TMR-DN ligand. **b** $R_g$ and $D_{max}$ of RhoBAST derived from guinier plots and PDDF calculation were plotted as a function of $Mg^{2+}$ concentration. Each data point represents an independent experiment ($n = 1$) and the error bars are propagated uncertainties calculated by GNOM. **c** Thermogram of the ITC experiment of RhoBAST binding to TMR-DN in the presence of 10 mM $Mg^{2+}$ (top) and overlay of integrated fitted heat plots in various $Mg^{2+}$ concentrations (bottom); for the arithmetic mean of the $K_D$ values and thermodynamic parameters, see Supplementary Table 7. **d** An overlay of the atomic model for wild-type RhoBAST alone (orange L2) built by FARFAR with the crystal structure of RhoBAST in complex with TMR-DN (gray L2) and U1A. For clarity, the U1A protein is not shown. **e–g** The theoretical scattering profile (shown as red solid line) calculated from the atomic model of RhoBAST WT was fitted with the experimental scattering profiles (shown as gray dots) of RhoBAST in the absence of $Mg^{2+}$ (**e**), in the presence of 0.5 mM (**f**), or both 10 mM $Mg^{2+}$ and TMR-DN (**g**). The error bars are propagated uncertainties calculated by GNOM. Source data for panels **a–c** and **e–g** are provided as a Source data file.

both A33G and A33C mutants have similar ligand binding affinity as wild-type RhoBAST. Correspondingly, their fluorescence activation ability roughly coincides with their ligand binding affinity.

We also investigated the nucleotides within the capping loop (Fig. 5b, c). While point mutants of G46A, G47A, U48C, and U49A decrease the ligand binding affinity by 6.2, 26.8, 1.3, and 100 folds, respectively, the mutant of G47C completely loses its ligand binding ability (Fig. 5b). Similar pattern is observed for their ability to activate TMR-DN fluorescence except for G46A, which enhances the fluorescence by 1.2 fold as compared to the WT RhoBAST (Fig. 5c). As G47 and U49 are more involved in the tertiary interaction networks than G46 and U48, these results underscore the importance of such interactions for the folding, ligand binding and concomitant fluorescence activation by RhoBAST.

Lastly, two-point mutants (G8G and G37A) and a deletion mutant (ΔU7) at the helices interface were analyzed (Fig. 5b, c). Residues of C8 and G37 hold the adjacent P2 and P4 helices together through a $Mg^{2+}$ clamp. The deletion mutant (ΔU7) is a mimic of the predecessor SRB-2 aptamer, in which the corresponding U7 is absent[19]. Such mutations are expected to affect the coaxial stacking of the helices as well as their relative orientation, which is presumed to be important for long-range tertiary interactions. However, none of these perturbations was found to prominently reduce the ligand binding affinity as well as fluorescence activation. Thus, the RhoBAST aptamer exhibits considerable structural resilience to perturbation around the helices interfaces.

## $Mg^{2+}$-dependence of RhoBAST folding and binding

It was previously reported that the fluorescence emission of TMR-DN in complex with RhoBAST highly depends on the concentration of $Mg^{2+}$[28]. To understand how $Mg^{2+}$ and ligand affect RhoBAST, we probe the tertiary folding and conformational changes of wild-type RhoBAST induced by $Mg^{2+}$ and ligand through SAXS. The scattering profiles, with scattering intensity $I(q)$ plotted against momentum transfer $q$, along with the pair distance distribution functions (PDDFs) transformed

from the scattering profiles of RhoBAST in the absence or presence of varying concentrations of $Mg^{2+}$ ($[Mg^{2+}]$: 0–10 mM) were shown in Supplementary Fig. 8. The dimensionless Kratky plot (Fig. 6a), plotted as $(qR_g)^2I(q)/I(O)$ vs $qR_g$, reflects the degree of flexibility and compactness of the molecule in solution. In the absence of $Mg^{2+}$, the RhoBAST displays a significant enrichment at high scattering angles, which is characteristic of a partially folded molecule in solution. As $Mg^{2+}$ increases, there are significant changes in the plots resulting in transitions to a more bell-shaped curve indicative of a folding event (Fig. 6a). The structural parameters derived from the PDDFs (Supplementary Table 6), including the radius of gyration ($R_g$) and the maximum particle dimension ($D_{max}$) which are indicators of the overall size of the molecule, were plotted as a function of $Mg^{2+}$ (Fig. 6b). Both $R_g$ and $D_{max}$ of RhoBAST decrease as $Mg^{2+}$ increases, consistent with a $Mg^{2+}$-induced structural transition from the unfolded to folded states. It's worth noting that though the RhoBAST RNA exhibits a partially folded structure in the absence of $Mg^{2+}$ (but with 100 mM $K^+$), a minimal concentration of $Mg^{2+}$ as low as 0.5 mM is sufficient to induce the proper folding of RhoBAST, of which the dimensionless Kratky plot is of high similarity as that in 10 mM, implying that RhoBAST folds efficiently at physiological concentrations of $Mg^{2+}$ (1–3 mM) and higher $Mg^{2+}$ causes little structural changes to the RNA.

To better understand how $Mg^{2+}$ affects the binding of RhoBAST to its ligand TMR-DN, ITC measurements were performed over $Mg^{2+}$ concentrations from 0 to 10 mM (Fig. 6c and Supplementary Table 7). While no TMR-DN binding can be observed in the absence of $Mg^{2+}$, TMR-DN binds to RhoBAST with moderate affinities at low $Mg^{2+}$ concentration (0.01 mM) and with a high affinity ($K_D$ ~ 24 nM) at 0.5 mM $Mg^{2+}$. However, further increasing of $Mg^{2+}$ concentration didn't confer the RNA with increased binding affinities. We also measured the fluorescence intensity of TMR-DN in the presence of an equal quantity of RhoBAST under different $Mg^{2+}$ concentrations. The RhoBAST–TMR-DN complex retains more than 90% of its maximum fluorescence under 0.1 mM $Mg^{2+}$. Together, these results imply that the binding of TMR-DN to RhoBAST is highly dependent on $Mg^{2+}$, and RhoBAST

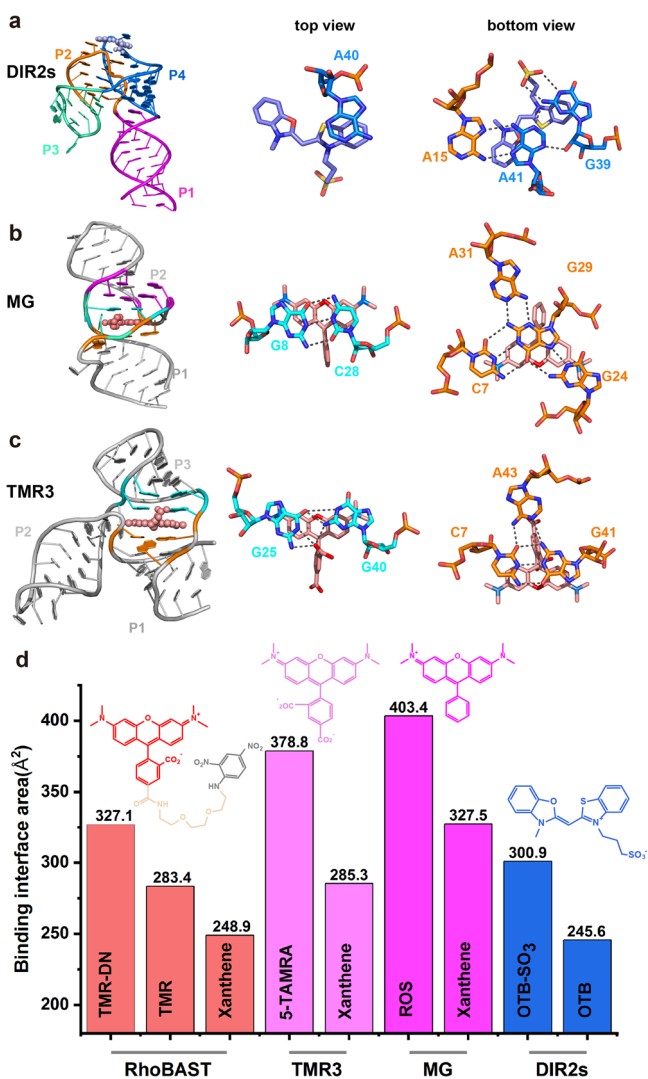

**Fig. 7 | Structural comparison with relevant aptamers. a** Cartoon representation of the three-dimensional structure of the DIR2s aptamer in complex with OTB–SO₃, and top and bottom views of the binding pockets with the ligand OTB–SO₃. **b** Cartoon representation of the three-dimensional structure of the Malachite green aptamer in complex with TMR, and top and bottom views of the binding pockets with the ligand TMR. **c** Cartoon representation of the three-dimensional structure of the TMR3 in complex with 5-TAMRA, and top and bottom views of the binding pockets with the ligand 5-TAMRA. In **a**–**c** the ligands were shown as spheres. **d** Comparison of the binding interface areas of RhoBAST, TMR3, MG, and DIR2s aptamers in complex with their cognate ligands and ligand moiety. The binding interface area value for each pair is indicated above the bar. Source data for panel (**d**) is provided as a Source data file.

functions effectively under physiological concentrations of Mg²⁺ (Supplementary Fig. 9).

To directly evaluate the crystal structure of RhoBAST and structural changes induced by Mg²⁺ and TMR-DN binding in solution, we built up an atomic model for wild-type RhoBAST by homology modeling using the program FARFAR2 and the crystal structure of the ternary complex and a canonical GAAA motif as template inputs[51] (Fig. 6d). We then compared the experimental SAXS profiles of RhoBAST WT under different conditions from that predicted based on the atomic structural model. While the experimental scattering profile of RhoBAST in the absence of Mg²⁺ significantly differs from its theoretical SAXS profile, the experimental scattering profiles of RhoBAST in the presence of 0.5 mM Mg²⁺, or both 10 mM Mg²⁺ and TMR-DN fit nicely with the theoretical SAXS profile (Fig. 6e–g). These results

indicate that Mg²⁺ is critical for proper folding of the binding pocket for efficient ligand binding. Even at low Mg²⁺ (such as 0.5 mM), the binding pocket is folded and exhibits strong structural rigidity and robustness, and subsequent ligand binding causes little structural changes in both the binding pocket and RNA.

## Structural comparison with relevant RNA aptamers

The efficiency of the FLAP-ligand system highly depends on the detailed characteristics of both the FLAP and its cognate fluorogenic ligand. Direct comparison of the structure and binding mode of relevant aptamer-ligand pairs is of significance to deepen our understanding of the underlying mechanisms and provide insights for rational design and optimization of both the aptamer and ligand.

We compared the crystal structures of ligand-bound RhoBAST and DIR2s aptamers[43] (Fig. 7a). Both DIR2s and RhoBAST are organized with a four-way junction, however, the two super-helices of H₂₃ and H₄₁ are roughly arranged in parallel in DIR2s, but antiparallel in RhoBAST, leading to extensive tertiary interaction networks between loops L2 and L4 in DIR2s and between loops L3 and L4 in RhoBAST (Fig. 7a). The ligand binding pockets in both DIR2s and RhoBAST have located at the apex rather the core of the structures. In DIR2s, the fluorophore plane of the ligand, oxazole thiazole blue (OTB) is sandwiched by the purine base triple A15–A41–G39 together with one unpaired nucleotide A40 from the top (Fig. 7a), yielding a binding interface area of 300.9 Å² which accounts for 52% of SASA of the ligand (Fig. 7d and Supplementary Fig. 4). The top surface of OTB plane is largely uncapped and exposed to the solvent, resembling the semi-open binding mode of TMR-DN with RhoBAST. Similar to RhoBAST, DIR2s exhibit considerable binding affinity to its cognate ligand (OTB) with $K_D$ ~ 700 nM. Additionally, the folding of DIR2s is ligand-independent, as evidenced by the crystallographic analysis that the structures of aptamer in apo- and holo-form are almost the same. While the folding and ligand binding affinity of both DIR2s and RhoBAST largely depend on Mg²⁺, at least 8 mM Mg²⁺ is required for DIR2s to achieve optimal binding capability, which is higher than the minimal Mg²⁺ concentration (0.25 mM) required in RhoBAST.

Next, we compared the structure of RhoBAST with that of two aptamers, the MG[37] and TMR3[45] that target rhodamine-based fluorophores (Fig. 7b, c). Different from RhoBAST, the MG and TMR3 aptamers are organized with a two-way and a three-way junction, respectively (Fig. 7b, c). Their ligand-binding pockets are located at the core rather than the apex of the structures. For ROS-bound MG aptamer, the xanthene plane of ROS is plugged between and stacked with two consecutive WC base pairs (C7:G29 and G8:C28) without apparent distortion of the base pairing, resulting in a continuous base stacking along with the two flanking helix stems (Fig. 7b). Interestingly, the Hoogsteen and sugar edges of the below base pair C7:G29 interact with the junctional residues of G24 and A31, respectively, and collectively form a base quadruple that constitutes the floor of the ligand binding pocket (Fig. 7b). Additionally, two A-minor base triples are stacked on the above G8:C28 base pairs. Apart from the stacking interactions with the xanthene moiety, the phenyl plane of TMR is stabilized by stacking with A30 and A9 (Fig. 7b). In the NMR structure of 5-TAMRA bound TMR3 aptamer, the ligand is sandwiched at the interface between P1 and P3 helices, leading to a continuous stacking on the xanthene moiety along with the closing base pair of P1 (C7:G41) and two consecutive noncanonical base pairs (G25:G40 and G26:A39) of P3 (Fig. 7c). These stacking interactions are likely to be reinforced by the A-minor interaction between A43 and C7:G41, and the hydrogen bonding interaction between the carboxylate group at *ortho*-position of phenyl ring and the amino groups of noncanonical base pair G25–G40 (Fig. 7c). These extensive intermolecular interaction networks in MG-ROS and TMR3-5-TAMRA pairs render binding interface areas of 378.8 Å² and 403.4 Å², accounting for the SASAs of ROS and 5-TAMRA by 72% and 60%, respectively, which are much higher than

that for TMR-DN in RhoBAST (48%) and OTB in DIR2s (52%) (Fig. 7d and Supplementary Fig. 4). However, no simple correlation between the binding interface areas and the binding affinity for MG and TMR3 aptamers can be seen. While the MG aptamer was found to have a higher binding affinity to TMR with a $K_D$ of around 40 nM[24], comparable to that of RhoBAST–TMR system (35 nM), TMR3 shows a modest binding affinity ($K_D$: 516 nM)[45].

## Discussion

In this work, we determined the crystal structure of the fluorescent aptamer RhoBAST in complex with TMR-DN, a contact-quenched fluorophore, and explored the conformational landscapes of free and RhoBAST-bound TMR-DN using MD simulations. We also studied the conformational dynamics of RhoBAST upon $Mg^{2+}$ and ligand binding by SAXS and ITC, of which the features are further compared with several related aptamer-ligand pairs in detail. Our results reveal a fluorogenic RNA fold and demonstrate the importance of tertiary interaction networks and $Mg^{2+}$ in dictating the architecture, folding, and dynamics of RhoBAST for efficient ligand binding and fluorescence activation.

We determined the high-resolution structure for FLAPs that exploit a CQ mechanism to activate its fluorogens. The RhoBAST structure demonstrates a fluorogenic RNA fold without employing a G4-based motif. Previously, the RhoBAST and its predecessor SRB-2 are predicted to form a three-way junction[28], the crystal structure reveals that RhoBAST adopts a four-way junctional topology instead. So far, high-resolution structures are available for up to 16 FLAPs (Supplementary Table 8). The majority of these FLAPs, such as the Spinach and its derivatives, Corn, Beetroot, the mango and its derivatives, consist of G-quadruplexes, or at least an isolated G-quartet motif in their fluorophore binding pockets[20,21,23,24]. Their cognate ligands, such as the analogs of *p*-HBI which is the intrinsic fluorophore of GFP, and the analogs of thiazole orange, contain twisted electron donor and electron acceptor moieties linked with a rotational bridge that allows for intramolecular rotation. The binding of fluorogens to these planar and large G4-based binding pockets restricts the intramolecular rotation, thus activating the fluorogens by the TICT mechanism.

Our results suggest strong structural robustness of RhoBAST in the presence of a physiological concentration of $Mg^{2+}$. SAXS and ITC data suggest that $Mg^{2+}$ is strictly required for the folding and high-affinity ligand binding of RhoBAST. RhoBAST adopts a partially unfolded conformation and is unable to bind with the ligand in the absence of $Mg^{2+}$, but the ligand binding affinities become stronger and the global shape of RhoBAST alone becomes almost the same as that in the crystal structure as the concentration of $Mg^{2+}$ increases up to 0.5 mM. These results support that the binding pocket is unfolded in the absence of $Mg^{2+}$ and ligand, and the reorganization of the binding pocket induced by $Mg^{2+}$ is a prerequisite for efficient ligand binding, similar as observed for several aptamer domains of riboswitches including the preQ1[52], the THF-II riboswitches[53]. However, in the presence of a physiological concentration of $Mg^{2+}$, the ligand binding pocket of RhoBAST is stably formed and subsequent ligand binding to RhoBAST only causes minimal structural perturbations, thus exhibiting strong structural robustness. These features are quite unusual since the folding of many aptamers including the MG aptamer are not only highly dependent on $Mg^{2+}$ but also subsequent ligand binding[37]. The recognition of ligands by RhoBAST resembles to the "lock-and-key" model. The structural rigidity could narrow the conformational space sampled by RhoBAST in the apo form, providing a rationale for its high-affinity ligand binding despite limited intermolecular interactions with the ligand.

The structural comparisons of RhoBAST with other related aptamer-ligand pairs disclose the structural basis for their different features in ligand recognition. The ligand binding pockets of RhoBAST and DIR2s located at the apex of the structure exhibit limited interactions with their respective ligands and the binding interfaces are relatively small (Fig. 7d). Thus it's not surprising the folding of RhoBAST and DIR2s aptamers is independent of ligand, and ligand binding only cause neglectable structural changes. By contrast, the ligand binding pockets of MG and TMR3 aptamers are located at the core of their structures, which establish extensive tertiary interactions between the aptamer and ligand and yield a larger buried interface upon ligand binding. Presumably, the ligand binding pockets in MG and TMR3 are structurally disordered in the absence of their cognate ligands[45], subsequent binding of ligands induces considerable reorganization of the global structure as well as ligand binding pocket. These features may explain the intricate balance between enthalpy and entropy for aptamer upon ligand binding, for example, the structural rigidity of RhoBAST and DIR2s aptamers compensates their limited interactions with the cognate ligand by means of reducing the entropic penalty upon binding (Supplementary Table 9). Furthermore, the rigid and semi-open ligand binding pockets in RhoBAST may endow a fast ligand exchange rate, allowing fast intermittency (blinking) in fluorescence emission which is a key requirement for single-molecule localization microscopy (SMLM). These features are expected to retain the RhoBAST–SpyRho and biRhoBAST–$TMR_2$ systems since their binding modes have been well reserved[29,32]. By contrast, for the Pepper systems, the binding or dissociation of ligands may require disruption or formation of tertiary interactions and structural reorganization, resulting in reduced ligand exchange kinetics which is at least two orders of magnitude slower than that of the RhoBAST–TMR-DN system[32], thus SMLM imaging using Pepper-HBC620 system was severely compromised because of its slow dye exchange and fast fluorescence decay[27].

Our results highlight the importance of tertiary interaction networks in dictating the folding, TMR-DN binding, and concomitant fluorescence activation of RhoBAST. It's believed that the strong association of RhoBAST to TMR-DN will drive the release of DN from TMR moiety, resulting in significant fluorescence unquenching. Any mutations that disrupt the folding or affect the ligand binding pocket may cause the loss of TMR-DN binding and in turn fluorescence activation. By contrast, the binding of RhoBAST to quencher-free TMR fluorophore remarkably boosts its quantum yield from 0.48 to 0.92[32]. The high quantum yield of the RhoBAST–TMR complex implies that RhoBAST binding efficiently activates the fluorescence of TMR through the TICT mechanism, and the quenching effects of RhoBAST nucleobases involved in interactions with TMR are relatively small. Similar small quenching effects are also expected to occur in the RhoBAST–TMR-DN complex. It's likely the quenching effects in RhoBAST–TMR-DN complex mainly come from the incomplete release of DN quencher from TMR moiety upon RNA binding, resulting in modest quantum yield for the complex, but not the quenching effects of nucleobases of RhoBAST. In support of this, there is a rough correlation between TMR-DN binding affinity and fluorescence activation ability among RhoBAST and its mutants (Fig. 5b, c). For residues that stack with the xanthene, such as G47 and A50, as these residues are either involved in the ligand binding pocket or directly interact with the ligand, any mutations result in reduced ligand binding affinity and fluorescence activation ability (Fig. 5b, c). However, there are also exceptions (G46A and A50U). Despite of its reduced binding affinity, the G46A mutant even slightly enhances fluorescence. As nucleobase 46 is adjacent to the dimethyl amino group of xanthene, such fluorescence enhancement effect could be attributed to the weaker quenching effect of adenine than guanine[54]. Regarding the A50U mutant, though it retains partial binding affinity with TMR-DN, it almost abolishes the fluorescence of the RhoBAST−TMR-DN complex. As A50 nucleobase is originally stacked with xanthene, the A50U mutation may destabilize the binding pocket and make it more flexible, in turn making the fluorophore xanthene more accessible to

quencher DN and resulting in an enhanced quenching effect. The quenching effects of RNA nucleobase on rhodamine-based fluorophores have also been observed in TMR3 aptamer, which quenches the fluorescence of its cognate ligand 5-TAMRA (similar to TMR compound) by 96% upon binding[45]. However, due to the complexity of RhoBAST structure, and the coexistence of photoexcitation and quenching processes, it is rather difficult to disentangle the role of single nucleotide or base-pair in fluorescence activation only by means of mutations.

Our structural and computational analysis also provides insights into the development and rational design of an improved TMR-DN-FLAP system. The close interaction between TMR and DN in TMR-DN alone is evidenced by its low quantum yield[15]. The notion has been widely held that fluorescence activation of TMR-DN requires releasing the DN quencher from TMR fluorophore upon RNA binding. While the crystal structure uncovered a StackP conformation in RhoBAST-bound TMR-DN, unexpectedly, our MD simulations suggest highly heterogeneous conformational ensembles with predominant contact-unstacked conformation for both free and RNA-bound TMR-DN. Interestingly, in-detail analysis and comparison of the conformational ensembles of free and RNA-bound TMR-DN showed that it's the fraction of StackX conformation in TMR-DN alone that significantly decreased from 28% to 1% upon RNA binding, by contrast, the DN-phenyl ring stacked conformation only slightly increased from 2% to 6%, thus suggesting the importance of DN-xanthene interaction in fluorescence quenching. The study may serve as a starting point for the rational design and optimization of both the aptamer and ligand of this important FLAP-ligand system. It's expected that the complete release of DN quencher from xanthene of TMR fluorophore will unquench TMR-DN more efficiently, resulting in even higher molecular brightness and high fluorescence quantum yield.

## Methods

### Small molecular fluorophore
Tetramethylrhodamine-dinitroaniline (TMR-DN) was synthesized as previously described[15]. Briefly, a solution of 5-carboxy-tetra-methylrhodamine-N-hydroxysuccinimide (1.0 mg, 1.9 μmol) in 100 μL of DMF was added to a solution of DN-PEG3-amine (1.8 mg, 5.7 μmol) dissolved in 50 μL of DMF. The reaction mixture was stirred at room temperature for 15 min, and purified on a reverse phase C-18 column (60% acetonitrile and 0.1% trifluoroacetic acid) to yield TMR-DN.

### RNA preparation
RNA constructs used in this study are listed in Supplementary Table 10. All the RNAs were prepared from PCR templates by in vitro transcription using homemade T7 RNA polymerase. To ensure the 3' homogeneity of the transcription product, two consecutive 2'-methoxy modifications were introduced to the 5' end of the reverse primers[55]. In vitro transcription was carried out at 37 °C for 2 h. The transcription supernatants were directly applied to a HiLoad 16/600 Superdex 75 gel filtration column and the RNAs were purified by size exclusion chromatography (SEC) and concentrated by centrifugal ultrafiltration (Amicon Ultra, 10 kDa MW cut-off). The SEC buffer contains 20 mM Tris-HCl (pH 7.5), 100 mM KCl, and 10 mM MgCl$_2$. Concentrated RNAs were stored at −80 °C until use. The concentrations of RNA were determined by UV-Vis absorbance at 260 nm using a NanoDrop 2000 (Thermo Scientific). The molar extinction coefficients of RNAs were calculated from the primary RNA sequences using the OligoAnalyzer Tool (https://sg.idtdna.com/calc/analyzer).

### Isothermal titration calorimetry
All the ITC measurements were performed with a Micro-Cal PEAQ-ITC microcalorimeter at 25 °C and the data were processed with the Origin7 software package from MicroCal. To test the effects of MgCl$_2$ on binding activity between RNAs and small molecular fluorophore,

the RhoBAST WT and mutant RNAs were exchanged into buffer containing 20 mM Tris-HCl (pH 7.5), 100 mM KCl, supplemented with different concentrations of MgCl$_2$ (0–10 mM) using SEC. About ∼280 μL of 10 μM RNA samples in each buffer were loaded into the sample cell. The syringe cell was filled with ∼45 μL of 100 μM small molecular fluorophore dissolved in the same buffer. The ligands were then titrated into the RNA solution with an initial 0.4 μL injection, followed by 19 serial 2 μL injections, with 90 s spacing time between each injection. The reference power was set as 10 μcal/s. The background data obtained from the buffer sample were subtracted before the data analysis. Integrated heat data were analyzed using the Origin7 software package provided by the manufacturer using a 'one set of sites' binding model. All the binding constants and thermodynamic parameters are listed in Supplementary Tables 5 and 7.

### Crystallization, diffraction data collection, and structure determination
The U1A−RhoBAST−TMR-DN cocrystals were grown by vapor diffusion at 16 °C. Sitting drops were prepared by mixing 1 μL of the RNA−fluorophore mixtures (150 μM RhoBAST−U1A RNA−protein complex and 200 μM fluorophore) with the same volume of a reservoir solution containing 0.2 M Sodium citrate tribasic dihydrate, 20% $w/v$ polyethylene glycol 3350. Rod-shaped crystals appeared within three days. Crystals were directly transferred to a solution consisting of 0.2 M Sodium citrate tribasic dihydrate, 20% $w/v$ polyethylene glycol 3350, and 20% ($v/v$) glycerol, immediately mounted in nylon loops (MiteGene) and flash-frozen by plunging in liquid nitrogen.

X-ray diffraction data were collected on beamline BL19U1 at the Shanghai Synchrotron Radiation Facility and indexed, integrated, merged, and scaled with HKL2000 (http://www.hkl-xray.com) or X-ray Detector Software[56]. The phasing of the crystal structure of U1A−RhoBAST−TMR-DN was determined by molecular replacement using the coordinates of U1A loop + U1A protein (PDB code: 5DDR) as a search model. All the models were built with the program COOT[57] and subsequently subjected to refinement by the program Phenix[58]. The data collection and crystallographic refinement statistics are listed in Supplementary Table 1. Structural figures were prepared using the program PyMOL and VMD 1.9.3.

The SASA was calculated using the VMD 1.9.3 program with a default solvent probe radius of 1.4 Å. The binding interface is defined by the equation: interface = 0.5 × (SASA$_{RNA}$ + SASA$_{ligand}$ − SASA$_{complex}$). The relative binding interface area is the ratio of the binding interface to SASA$_{ligand}$. It needs to be noted that in the case of aptamers containing the G4 motif, the potassium ions within the central channel of G4 motif were considered as a portion of RNA for SASA analysis, since the potassium ions are found to be essential for the formation of the G4 structure and directly interact with the ligands in some aptamers.

### Small-angle X-ray scattering
RNAs were prepared as described above, and fractionated by SEC (HiLoad 16/600 Superdex 75), the SEC buffer contains 20 mM Tris-HCl (pH 7.5), 100 mM KCl, 5 mM DTT, 3% glycerol, and different concentrations of MgCl$_2$ (0–10 mM). SAXS measurements were carried out at room temperature at the beamline 12 ID-B of the advanced photon source, Argonne National Laboratory. The setups were adjusted to achieve scattering $q$ values of $0.005 < q < 0.89$ Å$^{-1}$, where $q = (4\pi/\lambda)\sin(\theta)$, and $2\theta$ is the scattering angle. Thirty-two-dimensional images were recorded and reduced for each buffer or sample and no radiation damage was observed. Scattering profiles of the RNAs were calculated by subtracting the background buffer contribution from the sample buffer profile using the program PRIMUS3.2[59] following standard procedures. The two-dimensional images were reduced to one-dimensional scattering profiles using Matlab R2016b. The forward scattering intensity $I(0)$ and the radius of gyration ($R_g$) were calculated at low $q$ values in the range of $qR_g < 1.3$ using the Guinier

approximation. The $I(0)$ and $R_g$ were also estimated from the scattering profile with a broader $q$ range of 0.006–0.30 Å$^{-1}$ using the indirect Fourier transform method implemented in the program GNOM4.6[60], along with the PDDF, $p(r)$, and the maximum dimension of the protein, $D_{max}$. The parameter $D_{max}$ (the upper end of distance $r$) was chosen so that the resulting PDDF has a short, near zero-value tail to avoid underestimation of the molecular dimension and consequent distortion in low-resolution structural reconstruction. The volume-of-correlation ($V_c$) was calculated using the program Scatter and the molecular weights of solutes were calculated on a relative scale using the $R_g/V_c$ power law developed by Rambo et al.[61], independently of RNA concentration and with minimal user bias. The theoretical scattering intensity of the atomic structure model was calculated and fitted to the experimental scattering intensity using CRYSOL[62].

**Fluorescence enhancement assay**

Fluorescence scans were performed on an EnVision® Multi-mode Plate Reader set to excite and measure emission at wavelengths of interest. The fluorescence experiments for RhoBAST WT and its mutants were performed using the following conditions: 20 mM Tris-HCl (pH 7.5), 100 mM KCl, 10 mM MgCl$_2$, [RNA] = 10 μM, [TMR-DN] = 0.5 μM. To measure the Mg$^{2+}$-dependence of the fluorescence of TMR-DN alone and in the presence of an equal quantity of RhoBAST in the solvent, the fluorescence experiments were performed using the following conditions: 20 mM Tris-HCl (pH 7.5), 100 mM KCl, [RNA] = 10 μM, [TMR-DN] = 10 μM and various concentrations of MgCl$_2$ (1 μM–10 mM).

**Computational analysis of free and RhoBAST-bound TMR-DN**

We performed enhanced-sampling MD simulations to explore the conformational landscapes of free and RhoBAST-bound TMR-DN. The details regarding the quantum chemical calculations of TMR-DN alone, preparation of initial structural models for MD simulations, MD simulation control parameters, and analysis of MD simulation trajectories can be found in Supplementary Methods and Supplementary Fig. 10. To classify the resultant conformers in each ensemble from MD simulation, we analyzed the structural features including contact number, stacking interaction and the distance between centers of aromatic rings (centroid distance) in VMD using in-house scripts.

**Reporting summary**

Further information on research design is available in the Nature Portfolio Reporting Summary linked to this article.

## Data availability

The atomic coordinates and structure factors for the reported crystal structure have been deposited with the Protein Data Bank under the accession code 8JY0. We used the data deposited under PDB ID 5DDR to phase our crystal structure. The PDB files of other aptamers used for structural comparison are summarized in Supplementary Table 2 and are all accessible in the Protein Data Bank (https://www.rcsb.org). Source data are provided with this paper.

## Code availability

Custom scripts used to analyze the MD simulation data are provided with this paper in Supplementary Software 1.

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

## Acknowledgements

We thank the X-ray Crystallography Facility, Tsinghua University, and the beamlines of BL19U1 at the Shanghai Synchrotron Radiation Facility for providing technical support and assistance in X-ray crystallography data collection and analysis. We thank Ting Wang and the High Throughput Screening (HTS) Core Facility, Center of Pharmaceutical Technology, Tsinghua University for assistance with ITC measurements. This work was supported by grants from the National Key Research and Development Project of China (2021YFA1301500 to X.F. and 2023YFC2604300 to X.L.) and the National Natural Science Foundation of China (no. 82341081 to X.F. and no. 32271515 to X.L.).

## Author contributions

X.F. conceived and designed the project. Y.Z. prepared all the RNAs, performed the crystallization, data collection, and ITC measurements, and analyzed the SAXS data. Z.X. analyzed the structures and performed the computation. Y.X. determined the crystal structure. H.J. synthesized TMR-DN under the supervision of X.L., and X.Z. performed SAXS experiments. X.F., Z.X., and Y.Z. wrote the manuscript with inputs from the other authors.

 

## Competing interests

The authors declare no competing interests.
