## [Peer Review File · Nature Communications]

REVIEWER COMMENTS

Reviewer #1 (Remarks to the Author):

Zhang Y. et al. report the first co-crystal structure of RhoBAST RNA, a recently reported fluorescence aptamer in complex with a rhodamine derivative, and characterized the binding interface, fluorophore-quencher structural dynamics, Mg²⁺-dependency in RNA folding, fluorophore binding and fluorescence activation. This structure is entirely novel and surprisingly complex, and illustrates for the first time how RNA binding activates a contact-quenched fluorophore. The manner the Loops 3 and 4 interweave to form two long-range pairs, two base triples and to present fluorophore binding site is unique and quite remarkable. The work is very significant as it is a comprehensive analysis of a rare non-Quadruplex RNA activator of fluorescence, and its intricate structural organization provides important insights into complex RNA-RNA interactions and RNA-small molecule interactions. The work is technically sound, experiments well controlled, conclusions appropriate, and well written and clearly illustrated. I recommend this work for publication in Nat. Commun. with enthusiasm. I have some cosmetic suggestions below, which may help improve the clarity and general readability of the manuscript.

1. Line 49. "While enormous efforts have..." is logically clearer if "efforts" is changed to "progress".
2. Line 73. "well" should probably be replaced with "good".
3. Fig. 1d. I would suggest connecting the UUGG residues at the top with lines, so that the connectivity is clearer.
4. Fig. 3a. I would suggest adding a top down view of the ligand density. This side view is a bit hard to see clearly due to overlap.
5. Fig. 3f. The 90° rotation does not seem able to connect the two views.
6. Fig. 5a. I would suggest the authors annotate the mutations on the secondary structure such as in Fig. 1d. In the current secondary structure, it is very difficult to correlate the mutations with how they might impact the 3D structure or fluorophore binding.
7. Fig. 5b and legend. I suggest the authors use another symbol besides the asterisks, such as dagger, #, etc. Currently, the black versus red colors on the asterisks are the only distinguishing feature between the two marks. If one were to print the figure in black and white the two marks would look the same.
8. Fig. 5b. all the bars are similar in height but the folds of effect are in fact huge. I suggest the authors consider changing the lower bound of the y axis from zero to 3 or 4. This should spread out the differences and help visually distinguish among these bars.
9. Line 423. Why was homology modeling using FARFAR2 necessary here? Is the crystal structure plus a simple addition of the GAAA tetraloop in Coot not sufficient? It seems that the FARFAR2 model overlays nearly perfectly with the crystal structure except for the U1A-binding site anyway.

10. Line 430-434. Consider breaking this long and confusing sentence into two.

Reviewer #2 (Remarks to the Author):

Upon request, I will only comment the molecular dynamics part.

Basically, in my opinion, there are several concerns should be addressed for the MD part.

1. The objective to simulate the TMD-DN alone should be clarified.

I am not quite understand why the authors did not carry out dynamics simulation on the TMD-DN/RhoBAST complex. In fact, such kind of simulation, even in QM/MM level, is quite normal in computational biology field.

2. The authors claimed they employed MD for the generation of a conformational ensemble for TMD-DN alone. More importantly, they applied an implicit solvent model, and a semi-empirical method (XTB method) was employed for the QM part. My concern is if the author have verified the reliability of XTB on the geometric performance. Any higher level method for confirmation?

3. I do not quite think the TMD-DN (91 atoms) is a too large molecule to be calculated by DFT/MM method. Therefore, I would strongly suggest the authors at least carry out additional DFT/MM simulation with explicit water model for comparison.

4. For the choice of high-level DFT method, the authors selected an unusual used functional, r2scan-3c. In my opinion, more functionals should be tested here.

Reviewer #3 (Remarks to the Author):

Manuscript "Structural mechanism...by RhoBast" by Zhang et al. reports a novel co-crystal structure of the RhoBast RNA in complex with quencher tetramethylrhodamine-dinitroaniline (TMR-DN) where the DN quencher stacks over the phenyl group, away from xanthene group of TMR. MD calculations reveal the mode of self-quenching by a highly contact-but-unstacked conformational ensemble between TMR

and DN groups. In addition, the authors show the Mg^{2+} dependence of folding and binding of RhoBast. Overall, the study is well performed, and the main results are clearly illustrated. Because of the practical importance of FLAPs in RNA imaging, the work is a significant contribution to the field. The manuscript is appropriate for NatCom after major revisions.

Main concerns/recommended revisions

1. The authors revealed structural differences in fluorophore-quencher conjugate between the free and bound states by detailing interactions between the xanthene and DN, and phenyl ring and DN groups. These results explain the quantum yield difference between the free (quenched) and bound. However, the authors did not address the quenching effect by base rings of the residues that are stacked over xanthene. Those residues are U49, G47, A50 and A32 in the co-crystal structure.
2. Related to the previous question, the authors could have gone a step further to discuss the structural basis in terms of contact surfaces between the fluorophore group and FLAPs for the quantum yield difference between TMR-RhoBast complex and TMR alone, between TMR-RhoBast and TMR-DN-RhoBast complexes. This discussion might shed light on the quenching contributions from DN and stacking bases over the xanthene group of the TMR.
3. The authors appear largely focused on the discussion of the Mg^{2+} dependence of folding and binding affinity and avoided the discussion of the Mg^{2+} dependence of quenching. Furthermore, it is puzzling that the authors did not show the Mg^{2+} titration result of the fluorescence, which is critical for assessing the impact of the compactness in the binding pocket on fluorescence.
4. The authors made a number of mutations that are mostly aimed at probing folding. I'd recommend probing the binding and the quenching effects of the interface residues by mutating residues that stack over the xanthene group.

Minor edits

Overall, the manuscript is well illustrated and the results are well described. See the attached manuscript with trackmarked edits.

Response to reviewer 1

Reviewer #1 (Remarks to the Author):

Zhang Y. et al. report the first co-crystal structure of RhoBAST RNA, a recently reported fluorescence aptamer in complex with a rhodamine derivative, and characterized the binding interface, fluorophore-quencher structural dynamics, Mg²⁺-dependency in RNA folding, fluorophore binding and fluorescence activation. This structure is entirely novel and surprisingly complex, and illustrates for the first time how RNA binding activates a contact-quenched fluorophore. The manner the Loops 3 and 4 interweave to form two long-range pairs, two base triples and to present fluorophore binding site is unique and quite remarkable. The work is very significant as it is a comprehensive analysis of a rare non-Quadruplex RNA activator of fluorescence, and its intricate structural organization provides important insights into complex RNA-RNA interactions and RNA-small molecule interactions. The work is technically sound, experiments well controlled, conclusions appropriate, and well written and clearly illustrated. I recommend this work for publication in Nat, Commun. with enthusiasm. I have some cosmetic suggestions below, which may help improve the clarity and general readability of the manuscript.

Response #1: We thank the reviewer's for the positive comments and constructive suggestions on our work. We have addressed all the concerns point-by-point as below.

1. Line 49. "While enormous efforts have.." is logically clearer if "efforts" is changed to "progress".

Response #2: As suggested, we have replaced the "efforts" with "progress" in the revised manuscript.

2. Line 73. "well" should probably be replaced with "good".

Response #3: As suggested, we have changed the "well" with "good".

3. Fig. 1d. I would suggest connecting the UUGG residues at the top with lines, so that the connectivity is clearer.

Response #4: We connected the UUGG residues at the top with lines in Fig. 1d as suggested.

4. *Fig. 3a. I would suggest adding a top down view of the ligand density. This side view is a bit hard to see clearly due to overlap*

Response #5: As suggested, we have added a top down view of the ligand electron density map in Fig. 3a.

5. *Fig. 3f. The 90° rotation does not seem able to connect the two views.*

Response #6: Thanks for the reviewer's comment. We have replaced with a new 90° rotation view in Fig. 3f.

6. *Fig. 5a. I would suggest the authors annotate the mutations on the secondary structure such as in Fig. 1d. In the current secondary structure, it is very difficult to correlate the mutations with how they might impact the 3D structure or fluorophore binding.*

Response #7: In Fig. 5a, we annotated the mutations on the secondary structure shown in Fig. 1d as suggested.

7. *Fig. 5b and legend. I suggest the authors use another symbol besides the asterisks, such as dagger, #, etc. Currently, the black versus red colors on the asterisks are the only distinguishing feature between the two marks, If one were to print the figure in black and white the two marks would look the same.*

Response #8: As suggested, we have replaced the "red *" with "#" in Fig. 5b.

8. *Fig. 5b. all the bars are similar in height but the folds of effect are in fact huge. I suggest the authors consider changing the lower bound of the y axis from zero to 3 or 4. This should spread out the differences and help visually distinguish among these*

Response #9: Thanks for the reviewer's suggestion. We changed the lower bound of the y axis from zero to 4 as suggested.

9. *Line 423. Why was homology modeling using FARFAR2 necessary here? Is the crystal structure plus a simple addition of the GAAA tetraloop in Coot not sufficient? It*

seems that the FARFAR2 model overlays nearly perfectly with the crystal structure except for the UIA-binding site anyway.

Response #10: Both FARFAR2 and Coot can be used for homology modeling, but here we used FARFAR2.

10. Line 430-434. Consider breaking this long and confusing sentence into two.

Response #11: As suggested, we divided the original sentence into two sentences in the revised manuscript (Page 24).

Response to reviewer 2

Reviewer #2 (Remarks to the Author):

Upon request, I will only comment the molecular dynamics part.

Basically, in my opinion, there are several concerns should be addressed for the MD part.

1. The objective to simulate the TMD-DN alone should be clarified.

I am not quite understand why the authors did not carry out dynamics simulation on the TMD-DN/RhoBAST complex, In fact, such kind of simulation, even in QM/MM level, is quite normal in computational biology field.

Response #12: Thanks for the comments and suggestions. As the crystal structure only reveals the RNA-bound conformation of TMR-DN, and no experimental structural data is available for structure of TMR-DN alone. To better understand the fluorescence activation mechanism, we then simulated the conformation of TMR-DN alone. In the revision, we clarified the points in the beginning of the section regarding MD simulations. (2) In the revision, we performed enhanced sampling MD simulations on RhoBAST-bound TMR-DN with explicit water (OPC model) using AMBER/gaff2 force field (Fig. 4 in the main text, and Supplementary method 1). For comparison, we also performed MD simulation on TMR-DN alone (Fig. 4 in the main text) using the same force field as the complex, which is consistent with our previous MD simulation using xTB (GFN-FF potential functions rather than GFN-xTB) (ES Figure 1). Although TMR-DN comprise only 91 atoms, it contains 12 rotatable bonds between xanthene and DN moiety and exhibits rather structural heterogeneity, which could be highly challenging for QM/MM simulation, particularly for RhoBAST-TMR-DN complex system. Our computational analysis reveals that the DN-xanthene stacked conformers only account for minor portion (2% and 6%, respectively), but the contact-but-unstacked conformers are predominated in the ensembles of free and RNA-bound TMR-DN.

ES Figure 1. Comparison of population fractions for each subgroup of conformers from MD simulations on TMR-DN alone using GFN-FF/GBSA or gaff2/OPC model.

2. *The authors claimed they employed MD for the generation of a conformational ensemble for TMD-DN alone. More importantly they applied an implicit solvent model, and a semi-empirical method (XTB method) was employed for the QM part, my concern is if the author have verified the reliability of XTB on the geometric performance. Any higher level method for conformation?*

Response #13: We should make it more clear in our previous manuscript that we have employed force-field (GFN-FF) rather than semi-empirical method (GFN-xTB) to conduct MD simulation. In the revision, we chose the conformer, stackX-1, as our test sample, which is involved in lots of intramolecular interactions. We compared the geometry from XTB simulation with those from QM methods, including r2scan-3c method (previous manuscript), and PBE0-D3 functional combined with basis sets of increasing size including def2-TZVP, def2-TZVPP and def2-QZVP (ES Figure 2). The structures optimized with different QM methods here are almost indistinguishable. For convenience, we selected the structure optimized by r2scan-3c method as a reference. While the difference in bond angle between structure of GFN-FF and reference are rather small, the RMSD of bond-length and bond-angle involved in non-hydrogen atoms between them are 0.018 Å and 1.76 degree, respectively. And their non-hydrogen-atoms RMSD for phenyl and DN moieties is 1.67 Å, after alignment of xanthene ring. Similarly, we also verified the reliability of gaff2 on geometric performance. Compared to GFN-FF, gaff2 method seem to better reproduce the result of QM, as evidenced by the low RMSD value of 0.66 Å (vs 1.67 Å of GFN-FF). Nonetheless, these representative structures from MD simulations were

further re-optimized by QM method in our previous and revised manuscript.

ES Figure 2. Structural comparisons of the local minimal structures optimized with QM methods including r2scan-3c (red), PBE0-D3/def2-TZVP (tan), PBE0-D3/def2-TZVPP (blue) and PBE0-D3/def2-QZVP (cyan), as well as empirical force-fields including GFN-FF (silver) and gaff2 (gray). Given that the structures optimized with QM methods are almost indistinguishable, we separately visualized the structural differences between structures obtained by all QM methods (top) and between structures optimized by r2scan-3c and empirical force fields (bottom) for clarities. The structural alignments were conducted using the xanthene as reference. The geometric optimizations using QM methods or empirical force fields were performed with CPCM or GBSA implicit solvent models, respectively.

3. I do not quite think the TMD-DN (91 atoms) is a too large molecule to be calculated by DFT/MM method. Therefore, I would strongly suggest the authors at least carry out additional DFT/MM simulation with explicit water model for comparison.

Response #14: We agree with the reviewer that TMD-DN is not a too large molecule (91 atoms) to be modeled. However, TMR-DN contains 12 rotatable bonds between TMR and DN moieties, and exhibits highly dynamical structural feature, which could be highly computationally expensive if using QM with explicit water to explore conformational landscapes. To tackle the issues, we adopt multiple-scale approach, which comprises conformational exploration by classical MM simulation, geometrical optimization by medium level QM (r2scan-3c), and higher-level QM energy calculations (PWPB95-D3/def2-QZVPP, ω B97X-2-D3/def2-QZVPP).

4. For the choice of high-level DFT method, the authors selected an unusual used functional, r2scan-3c, In my opinion, more functionals should be tested here.

Response #15: As the reviewer suggested, we tested two other types of double-hybrid density functional combined with larger basis set, including PWPB95-D3/def2-QZVPP and ω B97X-2-D3/def2-QZVPP. The calculations using all three types of QM methods consistently support the notion that the representative conformers of contact conformation (i.e., stackX (X-1 and X-2), stackP, Contact-unstacked) share comparable relative energies (Supplementary Table 4). In our revised manuscript, we thus updated our energy calculation method.

Response to reviewer 3

Reviewer #3 (Remarks to the Author):

Manuscript "Structural mechanism...by RhoBast" by Zhang et al. reports a novel co-crystal structure of the RhoBast RNA in complex with quencher tetramethylrhodamine-dinitroaniline (TMR-DN) where the DN quencher stacks over the phenyl group, away from xanthene around of TMR, MD calculations reveal the mode of self-quenching by a highly contact-but-unstacked conformational ensemble between TMR and DN groups. In addition, the authors show the Mg²⁺ dependence of folding and binding of RhoBast. Overall, the study is well performed, and the main results are clearly illustrated. Because of the practical importance of FLAPs in RNA imaging, the work is a significant contribution to the field. The manuscript is appropriate for NatCom after major revisions.

Response #16: We appreciate the reviewer's positive comments and constructive suggestions on our work and have addressed all the major and minor concerns point-by-point as below.

Main concerns/recommended revisions

1. The authors revealed structural differences in fluorophore-quencher conjugate between the free and bound states by detailing interactions between the xanthene and DN, and phenyl ring and DN groups. These results explain the quantum yield difference between the free (quenched) and bound. However, the authors did not address the quenching effect by base rings of the residues that are stacked over xanthene, Those residues are U49, G47, A50 and A32 in the co-crystal structure.

Response #17: Thanks for the reviewer's comment. Inspired by the reviewer's comments, we discussed the important of these residues in dictating the folding, ligand binding and fluorescence activation of RhoBAST in the revision (Page 32-33). RhoBAST can bind with TMR-DN and quencher-free TRM with similar affinity. But the quantum yields of Rhobast-bound TMR-DN or TMR are 0.57 or 0.92, respectively. We feel like that quenching effects of RhoBAST-TMR-DN complex mainly come from the DN quencher, but not the nucleobases of RhoBAST. The residues that stack on xanthene, including G47 and U49, are involved in forming the

ligand binding pocket or directly interact with TMR, and any mutations of these residues may result in the loss of ligand binding and fluorescence activation abilities.

2. Related to the previous question, the authors could have gone a step further to discuss the structural basis in terms of contact surfaces between the fluorophore group and FLAPs for the quantum yield difference between TMR-RhoBast complex and TMR alone, between TMR-RhoBast and TMR-DN-RhoBast complexes. This discussion might shed light on the quenching contributions from DN and stacking bases over the xanthene group of the TMR.

Response #18: We appreciate the reviewer's comments and suggestions. As suggested, we added new discussions on the quenching contributions from DN and stacking bases over the xanthene group of the TMR in the revised manuscript (in Page 32-33, with track marked edits).

3. The authors appear largely focused on the discussion of the Mg^{2+} dependence of folding and binding affinity and avoided the discussion of the Mg^{2+} dependence of quenching. Furthermore, it is puzzling that the authors did not show the Mg^{2+} -titration result of the fluorescence, which is critical for assessing the impact of the compactness in the binding pocket on fluorescence.

Response #19: Thanks for the reviewer's suggestions. In the revised manuscript, we have discussed the effects of Mg^{2+} on fluorescence activation in detail (Page 23, the figure shown in SI Fig. 9). We found that at low concentrations of Mg^{2+} (< 0.1 mM), RhoBAST-TMR-DN complex can weakly activate the fluorescence, but it is far lower than that at increasing Mg^{2+} (≥ 0.1 mM). In line with this, Mg^{2+} affects the folding of RhoBAST itself and further affects the binding ability of RhoBAST with ligand.

4. The authors made a number of mutations that are mostly aimed at probing folding. I'd recommend probing the binding and the quenching effects of the interface residues by mutating residues that stack over the xanthene group.

Response #20: Thanks for the reviewer's suggestions. We performed mutational analysis for residues that stack over the xanthene group and discussed their effects in the revised manuscript (Page 33). Structural analysis suggested that the binding of

TMR-DN to RhoBAST was maintained by base stacking and hydrogen bonding, so if mutated, any disruption of these two kinds of interactions will affect the binding and activation ability of RhoBAST to TMR-DN.

Minor edits

Overall, the manuscript is well illustrated and the results are well described. See the attached manuscript with track marked edits.

Response #21: We appreciate the reviewer's suggestions, and we have made point-by-point modifications in the revised version based on these suggestions.

REVIEWERS' COMMENTS

Reviewer #1 (Remarks to the Author):

I think the authors have done a thorough and commendable job revising the manuscript and addressing the reviewer comments and suggestions. These revisions have substantially improved the manuscript. I recommend this work in its current form for publication in Nat. Commun. with enthusiasm.

Reviewer #2 (Remarks to the Author):

I am glad that the authors have addressed all my concerns. I'd like to recommend the manuscript to be accepted for publication.

Reviewer #3 (Remarks to the Author):

The authors have addressed all comments adequately.